# Psychometric Hepatic Encephalopathy Score for the Diagnosis of Minimal Hepatic Encephalopathy in Thai Cirrhotic Patients

**DOI:** 10.3390/jcm12020519

**Published:** 2023-01-08

**Authors:** Kessarin Thanapirom, Monton Wongwandee, Sirinporn Suksawatamnuay, Panarat Thaimai, Napaporn Siripon, Wanwisar Makhasen, Sombat Treeprasertsuk, Piyawat Komolmit

**Affiliations:** 1Division of Gastroenterology, Department of Medicine, Faculty of Medicine, Chulalongkorn University and King Chulalongkorn Memorial Hospital, Bangkok 10330, Thailand; 2Liver Fibrosis and Cirrhosis Research Unit, Chulalongkorn University, Bangkok 10330, Thailand; 3Excellence Center in Liver Diseases, King Chulalongkorn Memorial Hospital, Thai Red Cross Society, Bangkok 10330, Thailand; 4Department of Medicine, Faculty of Medicine, Srinakharinwirot University, Nakhon Nayok 26120, Thailand

**Keywords:** minimal hepatic encephalopathy, cirrhosis, psychometric hepatic encephalopathy score, animal naming test, Thai norms

## Abstract

The psychometric hepatic encephalopathy score (PHES) is the gold standard for diagnosing minimal hepatic encephalopathy (MHE). Screening for MHE is frequently overlooked in clinical practice due to time constraints. Furthermore, the simplified animal naming test (S-ANT1) is a new simple tool for evaluating MHE in cirrhotic patients. The purpose of this study was to standardize the PHES in a healthy Thai population, assess the prevalence of MHE, and validate the S-ANT1 in detecting MHE in patients with cirrhosis. The study included 194 healthy controls and 203 cirrhotic patients without overt HE. Psychometric tests and the S-ANT1 were administered to all participants. Multiple linear regression was used to analyze factors related to PHES results, and formulas were developed to predict the results for each PHES subtest. In healthy controls, age and education were predictors of all five subtests. The PHES of the control group was −0.26 ± 2.28 points, and the threshold for detecting MHE was set at ≤ −5 points. The cirrhotic group had PHES values of −2.6 ± 3.1 points. Moreover, MHE was found to be present in 26.6% of cirrhotic patients. S-ANT1 had a moderate positive correlation with PHES (r = 0.44, *p* < 0.001). S-ANT1 < 22 named animals detected MHE with a sensitivity of 71.2%, specificity of 65%, and area under the receiver operating curve of 0.68 (*p* < 0.001). In conclusion, Thai PHES normative data have been developed to detect MHE in cirrhotic patients who do not have overt HE. The optimal cutoff for detecting MHE in Thai cirrhotic patients was PHES ≤ −5 points and S-ANT1 < 22 named.

## 1. Introduction

Hepatic encephalopathy (HE) is a group of neuropsychiatric symptoms caused by liver insufficiency and/or portal-systemic shunting that range from mild alterations in brain function to marked disorientation and coma [1,2]. Minimal hepatic encephalopathy (MHE) is described as HE without obvious neurological abnormalities, but psychometric or neuropsychological testing reveals cognitive deficits [1,3,4]. MHE influences several cognitive domains, including attention and executive function, visuospatial perception, motor coordination, and psychomotor speed/reactions [3,5,6]. MHE is prevalent in up to 80% of cirrhotic patients [7,8,9,10] and is associated with decreased health-related quality of life, driving ability, and development of overt HE [8,11,12]. 

To detect MHE, several neuropsychological tests have been developed, including psychometric tests, electrophysiologic tests, and computerized tests, such as the critical flicker frequency and inhibitory control test [13,14,15,16]. A paper-and-pencil battery test called psychometric hepatic encephalopathy score (PHES) has been widely validated and recommended as the best clinical standard for diagnosing MHE [17,18,19]. The number connection tests (NCT) A and B, the digit symbol test (DST), the serial dotting test (SDT), and the line tracing test (LTT) all contribute to the PHES score. The PHES battery evaluates multiple areas of cognition related to the majority of neuropsychological deficits in MHE [5]. The tests are simple for medical personnel to perform and can be used across cultures. However, because the results of the PHES battery can be influenced by age, sex, and education level, each country should develop normative data based on its own cultural background [5]. Several countries, including Germany [5], Spain [20], Italy [21], the USA [22], China [23], and France [24], have standardized the PHES. However, no PHES standardization studies have been conducted in the Thai population.

Despite mounting evidence to the contrary, screening for MHE is frequently overlooked in routine practice due to the time required, the cost of testing, and the lack of available devices. As a result, a simple and low-cost test for MHE screening is required. The animal naming test (ANT) is a promising tool for evaluating cognitive function that has been validated in the diagnosis of MHE in cirrhotic patients [25,26,27,28]. Patients are challenged to name as many animals in one minute as they can. By adjusting for age and education, the simplified animal naming test (S-ANT1) is obtained. However, few studies have been conducted to assess the accuracy of S-ANT1 in diagnosing MHE in patients with cirrhosis. Therefore, this study aimed to (1) develop and validate normative data for the PHES in a healthy Thai population, (2) assess the applicability of PHES for detecting MHE among Thai patients with liver cirrhosis, and (3) validate S-ANT1 for determining MHE in patients with cirrhosis.

## 2. Materials and Methods

### 2.1. Participants

#### 2.1.1. Healthy Volunteers

The control group consisted of healthy volunteers aged 18–85 who visited King Chulalongkorn Memorial Hospital in Bangkok, Thailand, for a routine health checkup between May 2021 and April 2022. All subjects necessitated a basic knowledge of numbers and the Thai alphabetical sequence. For the control group, the following exclusion criteria were used: (1) the presence of chronic liver disease, neuropsychiatric disease, or other diseases that can impair cognitive function; (2) a prior history of chronic liver disease, neurologic, or psychiatric disorders; (3) use of psychoactive medications; (4) alcohol consumption greater than 50 g/day for males and 20 g/day for females; and (5) inability to read and write. 

#### 2.1.2. Patients with Cirrhosis

Cirrhotic patients from the Liver Clinic or hospital wards who did not have overt HE according to the West-Haven criteria [17,19] were consecutively enrolled in this study. Moreover, cirrhosis was diagnosed by radiological imaging. A physician (KT) conducted a neurological examination. Patients with overt HE according to the West-Haven criteria [17,19] or those with factors influencing neuropsychological status were excluded from the study. On the day of neuropsychological testing, demographic data, laboratory parameters, and Child-Turcotte-Pugh (CTP) classification were evaluated.

Each cirrhotic patient and healthy subject provided written informed consent. The local Institutional Review Board of Faculty Medicine, Chulalongkorn University (IRB No. 0171/65), approved the study protocol. The study protocol adhered to the Helsinki Declaration’s ethical principles and Good Clinical Practice guidelines. The study protocol was registered at the Thai Clinical Trial Registry (TCTR20221014005).

### 2.2. Psychometric Hepatic Encephalopathy Score

Five paper-pencil tests of the psychometric hepatic encephalopathy tests, including NCT-A, NCT-B, DST, SDT, and LTT (time and errors), were administered in the same order to all recruited healthy subjects and patients. Professor Karin Weissenborn kindly provided the PHES battery forms [5], which Dr. Monton Wongwandee translated into Thai. Due to incompatibilities between the German and Thai alphabets, the alphabet in the original version of NCT-B was replaced with the Thai alphabet in the same sequence. After a thorough explanation, demonstration, and training in a similar sequence of subtests, all subjects completed PHES. The tests were conducted one-on-one in a quiet room with sufficient lighting. The enrolled participants were supervised in completing these tests by a specially trained nurse (NS) and a research assistant (SS). 

All PHES tests were scored twice by two independent raters (KT and PT). If there was a disagreement when the data was gathered, the data were checked until a consensus was reached. The NCT-A, NCT-B, and SDT results were measured in seconds, including the time required to correct any errors, whereas the DST result was measured in corrected pairs. In LTT, results were calculated as the sum of complete time and error (LTTsum) [29]. The error points were assigned every time the drawn line touched (1 point) or crossed the boundary line (2 points) [29]. 

### 2.3. Animal Naming Test

Healthy subjects and patients with cirrhosis were asked to name as many animals as they could. All repetitions and mistakes were eliminated. The final score was the total number of named animals after 1 min. S-ANT1 was calculated by adjusting the effect of age and education on the ANT-1, as previously described [27]. Patients with less than 8 years of education received three animals, while those with less than 8 years of education but over the age of 80 received six. ANT1 and PHES were both tested on the same patient at the same time.

### 2.4. Statistical Analysis

Continuous variables were expressed as mean and standard deviation (SD), while categorical variables were expressed as a proportion. For continuous variables, the Mann-Whitney U-test was used, and for categorical variables, Pearson’s chi-square or Fisher’s exact test was used. Pearson’s and Spearman’s tests were used to examine correlation analyses. Each PHES subtest score was calculated using regression equations. The Kolmogorov-Smirnov test was used to determine whether the variables came from a normally distributed population. The sensitivity, specificity, positive and negative predictive value, area under the receiver operating characteristic (AUROC), and likelihood ratio of S-ANT1 for detecting MHE were all evaluated. By maximizing Youden’s index, the optimal threshold of S-ANT1 for differentiating MHE in cirrhotic patients was found. The reliability was analyzed by the Cohen’s kappa coefficient. SPSS software (version 22.0, IBM Corporation, Armonk, NY, USA) was used for statistical analysis. A two-sided *p* < 0.05 was considered statistically significant.

## 3. Results

### 3.1. PHES Standardization

This study included 194 healthy volunteers in order to create a normative database of PHES in Thais. The control group included 126 women (64.9%) with an average age of 47.1 ± 15.6 years (range 18–79 years). The average formal education duration was 13.8 ± 4.1 years (range 3–18 years). Table 1 summarizes the distribution of healthy volunteers classified by age group.

The results of NCT-A, NCT-B, SDT, LTT, and DST were 37.2 ± 8.4 s (s), 89.2 ± 14.2 s, 73.9 ± 13.3 s, 110.9 ± 12.5, and 45.9 ± 10.9 points, respectively. The results of all five tests were significantly correlated with age and education years. There was no relationship between gender and overall test performance. Pearson’s correlation between PHES test results and studied variables was shown in Table 2. 

For determination of the Thai norms for the PHES, the Kolmogorov-Smirnov test of normality revealed the normal distribution only of the DST. Other tests that did not conform to a normal distribution were transformed using logarithm (log) for NCT-A, NCT-B, SDT, and log-log for LTTsum. After transformation, these tests reached the normal distribution. The predictive equation for each subtest based on age and education level was constructed by multiple linear regression models (Table 3). The normal values were derived as the values of age-dependent mean and deviations of −1, +1, +2, +3 SDs for NCT-A, NCT-B, SDT, and LTTsum or +1, −1, −2, −3 SDs for DST from the mean value. Appendix A showed normal values with SDs based on age and education level (primary school, high school, and university) for each PHES sub-test. The final PHES was calculated by adding the results of five subtests with scores ranging from +5 to −15. The normative data of PHES was determined at the mean −2SD.

In the healthy subjects, the mean PHES score was −0.26 ± 2.28 (−8 to +10) points. The normal range of PHES was established at > −5 points. The pathological cutoff was determined to be mean −2SD. As a result, MHE was diagnosed when the score was ≤−5 points. The PHES score was significantly correlated with age (r =−0.62, *p* < 0.001) and education (r = 0.82, *p* < 0.001), but not with gender (r = 0.12, *p* = 0.09). Furthermore, no difference in scores was found between men and women (*p* = 0.11).

### 3.2. PHES Results in Patients with Cirrhosis

In total, 211 cirrhotic patients were screened for MHE. Eight patients were excluded from participating because they met at least one of the exclusion criteria. Finally, 203 patients without OHE were enrolled (age 57.5 ± 10.7 years, education 10.7 ± 5.1 years); 107 were male (52.7%); 89 (43.8%) had decompensated cirrhosis (Child-Pugh B/C). Cirrhosis was caused by hepatitis C virus (HCV) (*n* = 50, 24.6%), hepatitis B virus (HBV) (*n* = 50, 24.6%), non-alcoholic steatohepatitis (*n* = 38, 18.7%), alcoholic-related disease(*n* = 37, 18.2%), and other causes (*n* = 28, 13.8%). All HBV patients received anti-viral therapy at enrollment, and all HCV patients achieved sustained virological response at 12 weeks (SVR12) after direct-acting anti-viral treatment. Non-selective beta-blockers were administered to 59 (81.9%) patients, with no patients receiving antibiotic prophylaxis.

The results of NCT-A, NCT-B, SDT, LTTsum, and DST in cirrhotic patients were 59.8 ± 36.6 s, 137.8 ± 78.0 s, 99.7 ± 50.9 s, 148.7 ± 72.8, and 29.1 ± 12.7 points, respectively. The mean PHES score in patients with liver cirrhosis was −2.6 ± 3.1 points (median −2; range −14 to 4). The mean PHES in patients with cirrhosis was significantly lower than in healthy subjects (*p* < 0.001). Using a cutoff for MHE of ≤−5 points, 54 of 203 patients (26.6%) were diagnosed with MHE. Table 4 displayed PHES, individual test scores, and MHE-related variables in cirrhotic patients with and without MHE. Cirrhotic patients with MHE were older and less educated than healthy subjects. When compared to those without MHE, patients with MHE tended to have lower serum albumin (3.4 ± 0.7 vs. 3.6 ± 0.8 g/dL, *p* = 0.09) and higher proportion of CTP class B/C (53.7% vs. 40.3%, *p* = 0.08). MELD score, serum bilirubin and INR did not differ between patients with and without MHE. PHES had a weakly inverse relationship with CTP score (r = −0.19, *p* = 0.009), MELD score (r = −0.20, *p* = 0.008), serum bilirubin (r = −0.24, *p* = 0.001) and INR (r = −0.17, *p* = 0.02). Furthermore, PHES was also mildly positively correlated with serum albumin (r = 0.21, *p* = 0.005).

### 3.3. S-ANT1 in Cirrhotic Patients and Healthy Subjects and Its Performance in the Diagnosis of MHE

ANT1 was significantly correlated with age (r = −0.17, *p* = 0.02) and education level (r = 0.49, *p* < 0.001) in healthy subjects, but not with gender (r = −0.07, *p* = 0.31). S-ANT1 levels were higher in cirrhotic patients without MHE compared to those with MHE (24.5 ± 6.5 vs. 18.9 ± 6.5, *p* < 0.001). Furthermore, compared to cirrhotic patients with MHE, healthy subjects had higher S-ANT1 (26.8 ± 7.7) (*p* = 0.001). S-ANT1 had a moderately positive correlation with PHES score in cirrhotic patients (r = 0.44, *p* < 0.001). In a previous study, Campagna F et al. proposed a cutoff of S-ANT1 < 15 animals for MHE detection of MHE [27]. In the current study, using S-ANT1 < 15, names had 25% sensitivity, 96.5% specificity, 72.2% positive predictive value (PPV), 78% negative predictive value (NPV), and 0.61 AUROC (95%CI: 0.51–0.70, *p* = 0.02) for the diagnosis of MHE in cirrhotic patients. Moreover, the positive (+LR) and negative (-LR) likelihood ratio were 7.14 and 0.78, respectively.

The ideal cutoff for determining MHE in cirrhotic patients in the current study was S-ANT1 < 22. The analysis of Youden’s index with different S-ANT1 cut-offs was shown in Appendix A. Using this cutoff, the sensitivity was 71.2%, the specificity was 65%, the PPV was 42.5%, the NPV was 86.1%, the +LR was 2.03, the -LR was 0.44, and the AUROC was 0.68 (95%CI: 0.59–0.77, *p* < 0.001). Cohen’s kappa reliability measures showed a fair correlation between S-ANT1 < 22 and MHE diagnosed by PHES ≤ −5 points (k = 0.31, *p* < 0.001). MHE was detected in 25% and 71.2% of patients using S-ANT1 cutoff values of 15 and 22, respectively.

## 4. Discussion

The current study aimed to standardize normative data for the PHES in a healthy Thai population and to evaluate the PHES’s ability to detect MHE in patients with liver cirrhosis. In addition, we wanted to see how effective S-ANT1 was at detecting MHE. To our knowledge, this is the first study that provides PHES normative values in healthy Thai subjects. The main findings of this study are as follows: (i) the optimal cutoff of PHES for diagnosing MHE is ≤−5 points; (ii) PHES is influenced by age and education levels, not gender; thus, age and education-adjusted nomograms were established; (iii) MHE is present in 26.6% of cirrhotic patients without overt HE; and (iv) naming <22 animals best distinguished cirrhotic patients with and without MHE.

The PHES battery is a simple, sensitive, and low-cost tool for detecting MHE in patients with cirrhosis. The PHES normative data have previously been validated in several countries [5,10,13,23,24,30,31]. Although the basic structure of all PHES test versions is similar, there are significant differences in details that make comparing the results obtained from the various test versions challenging. Furthermore, age and education can have an impact on the PHES; normative data is required before using the PHES to diagnose MHE in populations with diverse cultural backgrounds [21,32]. A previous study from Germany [5] found that only age was related to PHES. The studies in Spain [20], Italy [21], France [24], Poland [33], China [23], and Korea [10] revealed that age and educational levels influence PHES. Consistent with the current study, we observed that the results of all subtests of PHES were affected by age and education. Therefore, the normalization of PHES was determined by the equation corrected with these two factors. The presence of MHE was associated with the severity of chronic liver disease.

In terms of the cutoff of PHES for determining MHE, a cutoff ≤−4 has been established in several countries, for instance, German [5], Italy [21], China [23], and Turkey [30], and has been recommended as national norms [34]. In contrast, studies in Polish [33], Indian [35], and Korean [10] cohorts showed that a PHES score ≤−5 was a diagnostic threshold for MHE. According to our findings, the PHES cutoff value in Thais is −5. The differences in details between test versions may explain why the results are discordant. First, the scoring systems for the LTT results and the range of total PHES scores were different. The German and Korean versions used two separate results (LTTerror and LTTtime) with a PHES score range of +6 to −18. The Italian, Chinese, Turkish, Polish, and the current versions used the sum (LTTsum) of time spent on the test plus error score with a PHES score range of +5 to −15. LTTsum was chosen for our study because it is practical, easy to implement in the clinic, and has been previously validated [29]. Second, the distribution and size of numbers and letters varied between versions. In addition, NCT-B was replaced with the figure connection test in Indian cohorts due to a large number of non-alphabetized patients [36]. Furthermore, the German alphabets in the NCT-B have been replaced by the alphabets of each country’s native language, such as Korean, Chinese, and Thai alphabets [10,23]. Third, normative data are obtained differently (age and education-adjusted values in Italy, China, Korea, Poland, and our study vs. age-adjusted values in Germany and India).

In our cohorts, MHE was found in 26.6% of cirrhotic patients. This finding is consistent with previous studies that found MHE prevalence ranging from 15% to 52.2% [10,23,30,35,37]. In contrast, the prevalence was high in the Cameroonian cohort (74%) [38]. Unlike previous findings [10,13,39], the study showed no association between Child-Pugh class and MHE. Consistent to some studies did not find this association [4,38,40]. The possible explanation might be from the Child-Pugh score could not be compared as a quantitative parameter, but rather as a qualitative variable separating two groups (CTP A vs. CTP B/C). The relationship between MELD and MHE remains controversial, with some studies indicating significant relationships [40] and others contradicting [41]. 

The present study validated the utility of S-ANT1 in detecting MHE. Campagna et al. proposed a three-level score of S-ANT1 for detecting no HE (≥15 animal names), MHE (<15 and ≥10 names), and overt HE (<10 names) [27]. S-ANT1 < 15 had a sensitivity of 78%, a specificity of 63%, a PPV of 61%, and an NPV of 79% for detecting impaired cognitive function. In contrast to our findings, this cutoff had low sensitivity (25%) and accuracy (AUROC 0.61) for detecting MHE [27]. Similar to Labenz C. et al., the study found that the sensitivity of S-ANT1 < 15 was 31% [25]. The differences in the characteristics of the enrolled patients between studies could be one explanation. Our and Labenz’s studies excluded patients with overt HE, but this was not performed in Campagna’s study. Patients with <8 years of education were found in 10.4% (*n* = 34/327), 1.4% (*n* = 2/143), and 34% (69/203) of the Campagna, the Labenz, and our studies. Using S-ANT1 < 22 named animals improved the sensitivity (71.2%) and accuracy (AUROC = 0.68) to detect MHE in our cohort. S-ANT1 was found to be significantly correlated with the PHES score. As a result, S-ANT1 is a simple and inexpensive tool for screening MHE in outpatient clinics or primary care centers.

This study has some limitations. The healthy controls were younger and more educated than the cirrhotic patients. Despite the fact that tests were standardized for age and education, these findings may have an impact on the threshold of PHES score for detecting MHE. Furthermore, the minority of the patients had decompensated cirrhosis with CTP B/C (35.5%), which may influence the prevalence of MHE in this population.

## 5. Conclusions

This study establishes Thai standards for the PHES test battery. For the diagnosis of MHE, a PHES cutoff of ≤−5 points is proposed. MHE was found in 26.6% of cirrhotic patients who did not have overt HE. MHE is more common in patients with severe liver disease. S-ANT1 < 22 is a simple and quick test that can help physicians evaluate MHE, especially in primary care hospitals.

## Figures and Tables

**Table 1 jcm-12-00519-t001:** Distribution of healthy volunteers according to age group.

	Gender (Male/Female)	Education (Years)
18–30 years (*n* = 31)	14 (45.2%)/17 (54.8%)	15.5 ± 2.3
30–40 years (*n* = 40)	12 (30%)/28 (70%)	14.4 ± 3.2
40–50 years (*n* = 30)	12 (40%)/18 (60%)	14.4 ± 3.9
50–60 years (*n* = 47)	16 (34%)/31 (66%)	13.6 ± 4.3
60–79 years (*n* = 46)	14 (30.4%)/32 (69.6%)	11.9 ± 5.1

**Table 2 jcm-12-00519-t002:** Correlation between the results of PHES and studied factors in healthy subjects.

	Age	Education Years	Gender
NCT-A	r = 0.32, *p* < 0.001	r = −0.44, *p* < 0.001	r = −0.12, *p* = 0.11
NCT-B	r = 0.38, *p* < 0.001	r = −0.33, *p* < 0.001	r = 0.04, *p* = 0.55
SDT	r = 0.56, *p* < 0.001	r = −0.38, *p* < 0.001	r = 0.01, *p* = 0.99
LTT	r = 0.24, *p* < 0.001	r = −0.25, *p* < 0.001	r = 0.03, *p* = 0.69
DST	r = −0.32, *p* < 0.001	r = 0.56, *p* < 0.001	r = −0.04, *p* = 0.55

NCT-A, number connection test-A; NCT-B, number connection test-B; SDT, serial dotting test; LTT, line tracing test; DST, digit symbol test; PHES, psychometric hepatic encephalopathy score.

**Table 3 jcm-12-00519-t003:** Predictive equations of psychometric hepatic encephalopathy score of each sub-test.

Test	Equation	SD
Log (NCT-A)	1.524 + (0.004 × age) − (0.014 × education year)	0.15
Log (NCT-B)	1.952 + (0.002 × age) − (0.011 × education year)	0.17
Log (SDT)	1.926 + (0.002 × age) − (0.014 × education year)	0.15
Log-log (LTT-sum)	0.310 + (0.00032 × age) − (0.001 × education year)	0.03
DST	49.561 − (0.492 × age) + (1.412 × education year)	9.7

NCT-A, number connection test-A; NCT-B, number connection test-B; SDT, serial dotting test; SD, standard deviation; LTT, line tracing test; DST, digit symbol test.

**Table 4 jcm-12-00519-t004:** PHES and individual test scores and variables related to MHE in cirrhotic patients with and without MHE.

	No MHE (*n* = 149)	MHE (*n* = 54)	*p*-Value
Age, years	56.3 ± 12.3	60.6 ± 13.4	0.02
Male, *n* (%)	81 (54.4%)	26 (58.1%)	0.43
Education level, years	11.3 ± 5.0	9.1 ± 5.1	0.01
Cause of cirrhosis, *n* (%)			
Chronic HCV	32 (21.5%)	18 (33.3%)	0.22
Chronic HBV	40 (26.8%)	10 (18.5%)	
NASH	28 (18.8%)	10 (18.5%)	
CTP, *n* (%)			
A	89 (59.7%)	25 (46.3%)	0.09
B/C	60 (40.3%)	29 (53.7%)	
MELD	11.6 ± 4.8	12.7 ± 6.6	0.54
Total bilirubin, mg/dL	1.8 ± 1.9	3.9 ± 8.7	0.93
Albumin, g/dL	3.6 ± 0.8	3.4 ± 0.7	0.06
INR	1.3 ± 0.3	1.5 ± 1.2	0.57
NCT-A, seconds	48.7 ± 21.1	90.8 ± 50.2	<0.001
NCT-B, seconds	113.8 ± 51.0	202.3 ± 99.1	<0.001
SDT, seconds	83.4 ± 29.3	146.2 ± 66.1	<0.001
LTTsum	128.7 ± 48.0	203.8 ± 97.7	<0.001
DST, points	32.5 ± 12.3	19.4 ± 8.1	<0.001
PHES	−1.2 ± 1.9	−6.7 ± 2.0	<0.001

CTP, Child-Turcotte-Pugh classification; DST, digit symbol test; HBV, hepatitis B virus; HCV, hepatitis C virus; LTTsum, line tracing test (error + time); NASH, non-alcoholic steatohepatitis; NCT-A, Number connection test-A; NCT-B, number connection test-B; PHES, psychometric hepatic encephalopathy score; SDT, serial dotting test.

## Data Availability

The authors will make available the anonymized data used in this work upon request.

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
