# Peer review of "Psychometric Hepatic Encephalopathy Score for the Diagnosis of Minimal Hepatic Encephalopathy in Thai Cirrhotic Patients"

_jcm, 2023, doi:10.3390/jcm12020519_

Round 1

Reviewer 1 Report

Dear authors,

I read your manuscript with interest and I fully appreciate that this type of research is neccessary for clinical practice, but quite often it is difficult to publish.

The manuscript aims to provide normal values for PHES battery of tests for Thai population derived from a cohort of healthy volunteers and applied on a cohort of patients with liver cirrhosis.

Major comments

1) manuscript requires a straightforward explanation how the normal values were derived. Are they derived directly from the observed cohort + 2SD, or the observed cohort was used to construct a model and normal values were derived from expected values adjusted for age and education +2SD (I would suggest the latter)

2) to derive a cut off authors should describe distribution of the variable - are the results of PHES tests from normal distribution? if not what has been done to adress this issue?

3) authors suggest that the results of PHES battery of tests were significantly correlated with age and education (expressed as years of schooling). Authors even provide regression equations for these variables, however this is not reflected in the recommended cut-off,  the authors provide only the normal values for the whole population. Even in the manual that accompanies PHES battery from Uni Hannover, normal values are provided for every age. (this should be reflected in the results  - normal values for age and education categories) - maybe in the supplementary data.

4) Internal testing consistency should be also reported by test-retest correlation in healthy population although this is not mandatory

5) I suggest that the main findings - suggested cut-offs for PHES battery and its components as well as the ANT cut-off for covert HE are included in the abstract.

6) Regarding the validation of PHES suggested in the conclusions - PHES battery has been validated multiple times against quantitative EEG. As qEEG is independent from cultural background this "criterion" validation is not neccessary in adapting the PHES to different population (contrary to the statement in the abstract)

7) Authors also aimed to validate ANT test. Here the criterion validation against PHES could be carried out, however the statisticall procedure should take into account that this is indeed a repeated measures procedure, thus gennerally does not fulfill the assumptions of either correlation or regression analysis. This should be done by intraclass correlation or Cohens kappa.

Reviewer 2 Report

 Is a very interesting study; however some concerns should be answer by the authors:

1-       In the table 4 is not clear, in the etiologies, the viral, if those patients included are under treatment or without treatment? Is different the probability of decompensation in both groups, and is something to consider.

2-      Another very important issue, most of the included patients are compensated cirrhotic patients, with good liver function a low risk of hepatic encephalopathy. How the authors can explain the main results? Is a big limitation in this study.

3-      In the decompensated patients or those with worse liver function, is not clear the chronic medication, neither how many of them are under treatment with NSBBs neither prophylaxis with antibiotics.

Round 2

Reviewer 1 Report

my comments from the first review have been sufficiently addressed, I recommend publication as is.

Reviewer 2 Report

All my comments have been taken into account in the current version of the manuscript.